Host-Microbe Biology
# Physical Activity Shapes the Intestinal Microbiome and Immunity of Healthy Mice but Has No Protective Effects against Colitis in MUC2−/− Mice

Mehrbod Estaki,[a,b] Douglas W. Morck,[c] S. Ghosh,[a] Candice Quin,[a] Jason Pither,[a] Jacqueline A. Barnett,[a] Sandeep K. Gill,[a,d] Deanna L. Gibson[a]

[a]Department of Biology, Faculty of Science, University of British Columbia, Kelowna, British Columbia, Canada
[b]Department of Pediatrics, University of California San Diego, La Jolla, California, USA
[c]Department of Biological Sciences, University of Calgary, Calgary, Alberta, Canada
[d]School of Population and Public Health, University of British Columbia, Vancouver, British Columbia, Canada

**ABSTRACT** The interactions among humans, their environment, and the trillions of microbes residing within the human intestinal tract form a tripartite relationship that is fundamental to the overall health of the host. Disruptions in the delicate balance between the intestinal microbiota and host immunity are implicated in various chronic diseases, including inflammatory bowel disease (IBD). There is no known cure for IBD; therefore, novel therapeutics targeting prevention and symptom management are of great interest. Recently, physical activity in healthy mice was shown to be protective against chemically induced colitis; however, the benefits of physical activity during or following disease onset are not known. In this study, we examine whether voluntary wheel running is protective against primary disease symptoms in a mucin 2-deficient (*Muc2−/−*) lifelong model of murine colitis. We show that 6 weeks of wheel running in healthy C57BL/6 mice leads to distinct changes in fecal bacteriome, increased butyrate production, and modulation in colonic gene expression of various cytokines, suggesting an overall primed anti-inflammatory state. However, these physical activity-derived benefits are not present in *Muc2−/−* mice harboring a dysfunctional mucosal layer from birth, ultimately showing no improvements in clinical signs. We extrapolate from our findings that while physical activity in healthy individuals may be an important preventative measure against IBD, for those with a compromised intestinal mucosa, a commonality in IBD patients, these benefits are lost.

**IMPORTANCE** Perturbation in the gut microbial ecosystem has been associated with various diseases, including inflammatory bowel disease. Habitual physical activity, through its ability to modulate the gut microbiome, has recently been shown to prophylactically protect against chemically induced models of murine colitis. Here, we (i) confirm previous reports that physical activity has limited but significant effects on the gut microbiome of mice and (ii) show that such changes are associated with anti-inflammatory states in the gut, such as increased production of beneficial short-chain fatty acids and lower levels of proinflammatory immune markers implicated in human colitis; however, we also show that (iii) these physical activity-derived benefits are completely lost in the absence of a healthy intestinal mucus layer, a hallmark phenotype of human colitis.

**KEYWORDS** IBD, colitis, exercise, inflammatory bowel disease, microbiome, mucin, physical activity, SCFA, short-chain fatty acids

Address correspondence to Deanna L. Gibson, deanna.gibson@ubc.ca.

Exercise has been shown to alter the gut microbiome, SCFA, and be protective in chemically-induced models of acute colitis. However, in the absence of a normal mucosal layer, all of these benefits of exercise on gut health are lost.

[This article was published on 6 October 2020 but required additional changes, now reflected in the Note Added after Publication on p. 18. The changes to the article were made on 27 October 2020.]

Inflammatory bowel diseases (IBD), encompassing Crohn's disease (CD) and ulcerative colitis (UC), are idiopathic, relapsing chronic diseases characterized by chronic inflammation of the gastrointestinal tract. While pathology varies between UC and CD, both

burden patients with common debilitating clinical symptoms, such as diarrhea, rectal bleeding, abdominal pain, and weight loss. The etiology of IBD is not known; however, a combination of genetic, immunological, and environmental factors is implicated in its development. Most recently, the contribution of the intestinal microbiota in IBD pathogenesis has arisen as an active area of research (1). For example, IBD patients have reduced gut microbial diversity (2) and are more likely to have been exposed to antibiotics in the 2 to 5 years preceding their diagnosis (3). In animal models, mice genetically predisposed to colitis (IL-10$^{-/-}$) are resistant to disease onset while being kept under germfree conditions, but clinical signs instigate immediately following exposure to microbes (4).

With the incidence of IBD and its burdens rising globally (5), there is an increasing demand for novel therapeutics. Physical activity (PA) has been proposed as both a primary and an adjunct therapy for the prevention and treatment of various chronic diseases due to its well-documented ability to ameliorate low-grade systemic inflammation (6). Most recently, IBD has been marked as a potential new candidate (7) that can yield benefits from regular PA. Studies of PA in rodents have shown attenuated clinical signs of chemically induced colitis (8–11) that appear to be dependent on the colitis model and type of PA. These studies, however, only assess the role of PA as a preventive measure leading up to the induction of acute colitis via a chemical toxin. As such, the potential benefits of PA succeeding or during disease onset is not known. In this study, we aimed to address this knowledge gap by utilizing the mucin 2 knockout ($Muc2^{-/-}$) mouse model of chronic colitis.

The human intestinal tract is continuously exposed to the trillions of microbes residing within the mucosal layer of the lumen. Under homeostatic conditions, these microbes are tolerated by the host, as they provide essential functions, such as digestion of complex carbohydrates, protection against enteric pathogens, and production of beneficial short-chain fatty acids (SCFA), to name a few. Separating the luminal microbes from intestinal epithelial cells (IEC) is a mucus bilayer largely composed of the highly glycosylated protein MUC2. In the colon, the loosely structured outer mucus layer allows for colonization of microbes in a nutrient-rich environment, while the dense inner layer segregates them from the IEC (12). Inflamed intestinal tissues of UC patients commonly display structural defects or thinning of this mucus layer (13), leading to excessive exposure of microbial antigens to the host cells. This prompts a chronic state of inflammation and apoptosis, leading to further loss of IEC integrity and, thus, further exposure and injury. $Muc2^{-/-}$ mice or those with missense mutations impairing the release of MUC2 are born with an underlying predisposition to intestinal inflammation that shows the rapid progression of colitis (14). $Muc2^{-/-}$ mice generally display early clinical signs of colitis following weaning (~1 month) and microscopic tissue pathology at as early as 2 months of age, indicating moderate-level colitis, reaching high severity by 4 months (15).

We hypothesized that the introduction of $Muc2^{-/-}$ mice to voluntary wheel running (VWR) immediately following weaning would reduce the severity and delay the onset of clinical signs of colitis. Having recently shown a significant correlation between aerobic fitness and overall microbial diversity and increased butyrate production (16), we hypothesized further that PA-associated protection would be mediated through changes of the intestinal microbiota and their metabolites.

## RESULTS

**Wheel running is comparable between WT and $Muc2^{-/-}$ mice.** For unknown reasons, one animal from each group did not run on the wheels and was excluded from further analyses. The wild-type (WT) group ran an average (standard deviation [SD]) of 46.6 (18.4) km in total throughout the 6 weeks, while the $Muc2^{-/-}$ animals ran slightly less, at 40.7 km (21.5), which corresponds to ~1.3 and 1.1 km/day, respectively. While the WT showed a general trend toward more wheel running, the differences were not statistically significant (Fig. 1A), likely due to the highly variable nature of running data.

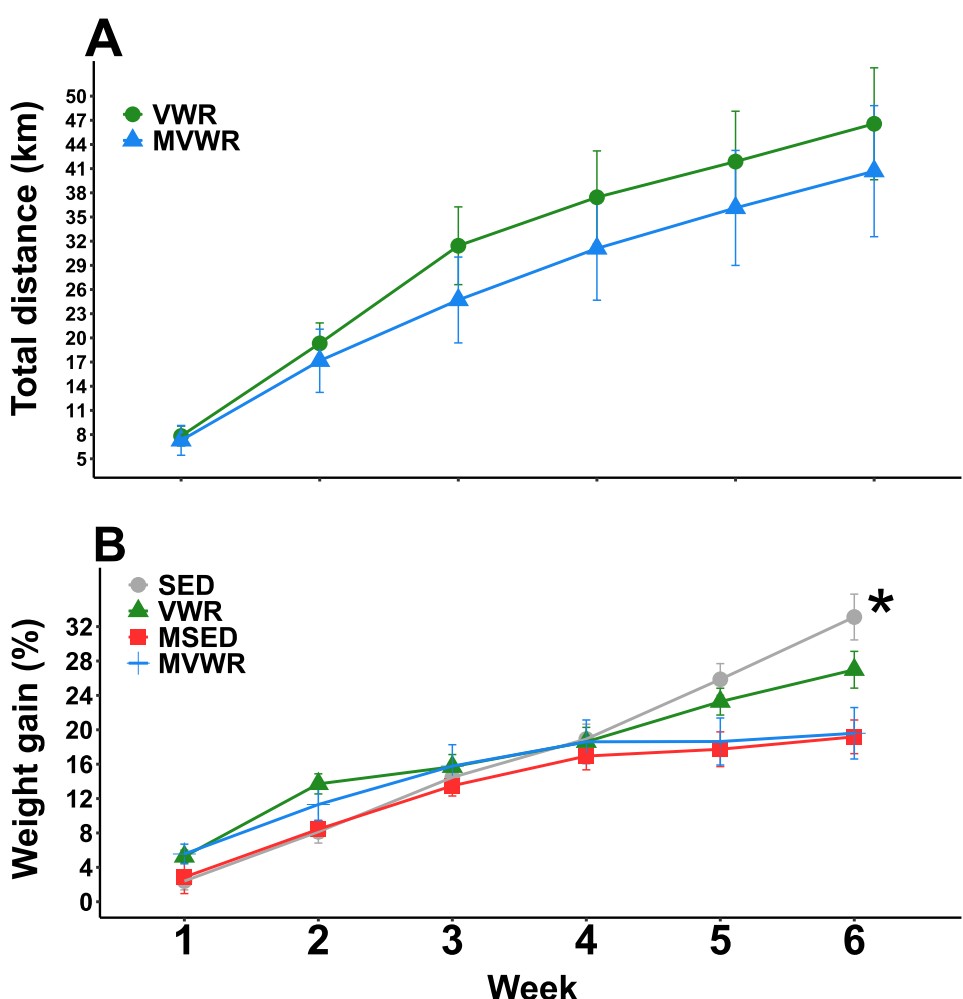

**FIG 1** Weekly measures of relative weight gain and wheel running. Longitudinal measurements of average accumulated distance ran (A) and relative weight gain compared to week 0 (B). Linear mixed models were used with week and animals set as random effects. There were no significant effects of wheel running in either wheel running or weight gain. An asterisk indicates a significant ($P < 0.05$) main effect between genotypes.

***Muc2*$^{-/-}$ mice gained less weight than WT mice despite similar food intake.** Weight gain was not significantly different across activity levels; however, as expected, *Muc2*$^{-/-}$ mice gained less weight throughout the 6 weeks (Fig. 1B). The mean ($\pm$ standard errors [SE]) total weight gain of each group, relative to their starting body weights, was the following: mice on a locked wheel (SED), 33.12% $\pm$ 2.66%; VWR, 26.98% $\pm$ 2.14%; *Muc2*$^{-/-}$ mice with access to a locked wheel (MSED), 19.18% $\pm$ 1.96%; and *Muc2*$^{-/-}$ mice with access to a free wheel (MVWR), 19.59% $\pm$ 2.99%. By the final week, VWR animals had gained ~6% less total weight than their SED counterparts ($P = 0.09$). Food intake was not statistically different between groups across the 6 weeks (see Fig. S1A in the supplemental material). *Muc2*$^{-/-}$ mice drank significantly more water than WT animals (beta coefficient [B], 5.4; $P < 0.001$) throughout the 6 weeks. Wheel running was associated with increased water intake in WT (B, 5.3; $P < 0.01$) and, to a lesser extent, in *Muc2*$^{-/-}$ (B, 1.9; $P < 0.86$) mice (Fig. S1B).

**Wheel running did not reduce histopathological or clinical scores in *Muc2*$^{-/-}$ mice.** We found a modest but significant difference in disease score between MVWR and MSED groups (B, $-1.67$; $P < 0.01$) across the 6 weeks, indicating reduced clinical signs in the running animals. However, *post hoc* tests carried out each week showed no significant difference between groups (Fig. 2A). The differences between groups appear to increase with time, with the largest difference appearing at week 6 (B, $-2.0$;

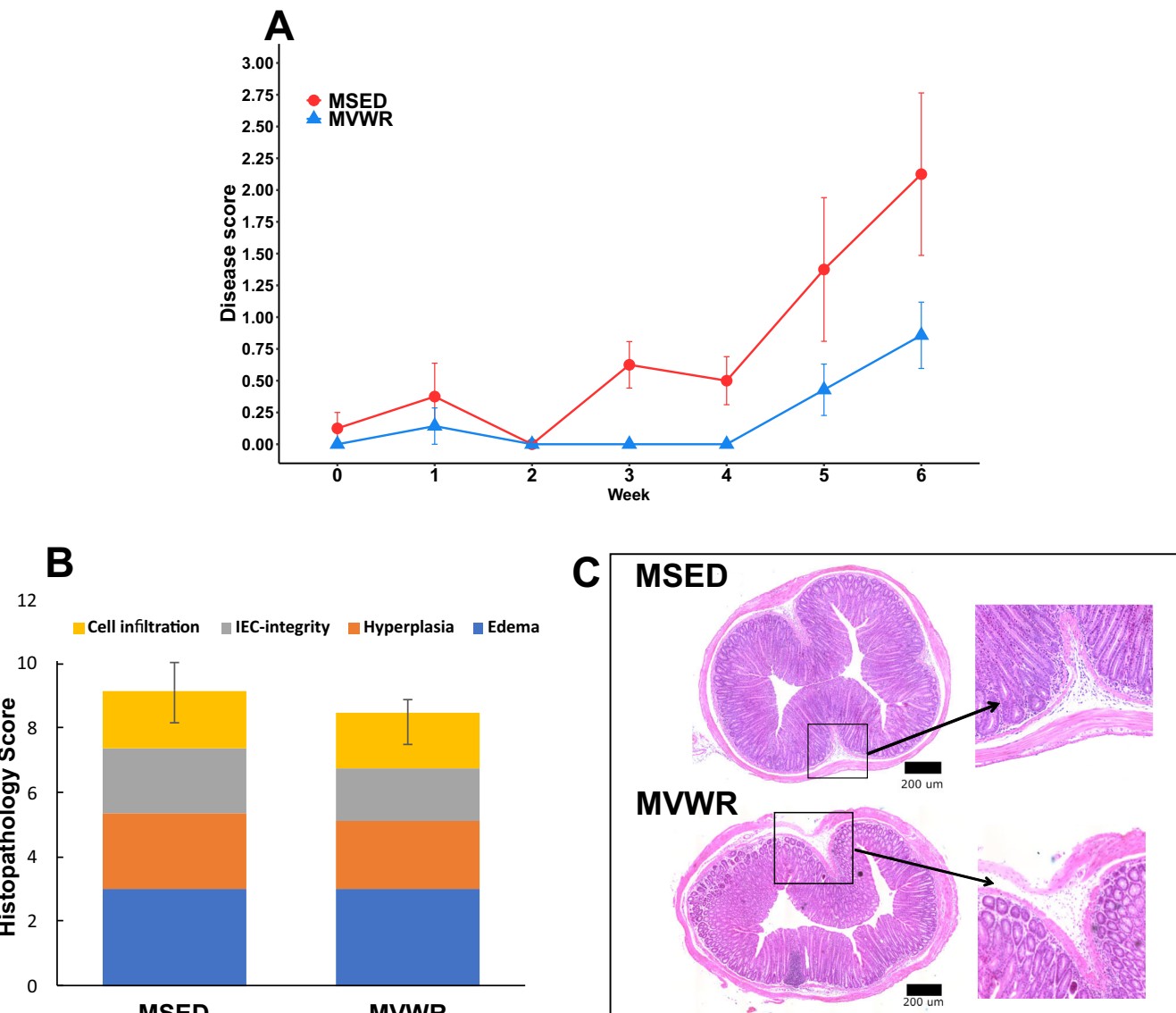

**FIG 2** Assessment of severity of colitis signs in *Muc2*−/− mice. (A and B) Comparison of clinical disease scores across 6 weeks (A) and histopathological scores in *Muc2*−/− animals at the terminus (B). (C) Representative colon images of H&E-stained sections from MSED and MVWR mice. No significant differences were observed between groups in either measurement. Values are shown as means ± SE.

*P* = 0.063). Histopathological scores based on hematoxylin and eosin (H&E)-stained sections showed no differences among the groups (Fig. 2B).

**Wheel running altered short-chain fatty acids in WT but not *Muc2*−/− mice.** There was a significant group effect (beta coefficient, 14.83; *P* < 0.01) in SCFA abundances. Specifically, a significant difference in total SCFA (Fig. 3), acetate, propionate, butyrate, and valerate was detected across groups. The total SCFA concentration was significantly higher in VWR mice than in all other groups, while SED mice had total SCFA similar to both *Muc2*−/− groups. VWR mice also had significantly higher total acetate and butyrate than all the other groups and higher levels of propionate than SED. Overall, the major difference between *Muc2*−/− and WT animals was the significantly reduced levels of butyrate in *Muc2*−/− mice and, inversely, higher levels of propionate. Valerate, caproate, and heptanoate were similar across all groups. In terms of relative abundances, the main differences between *Muc2*−/− and WT mice were the higher propionate and lower butyrate proportions in *Muc2*−/− animals. Importantly, the

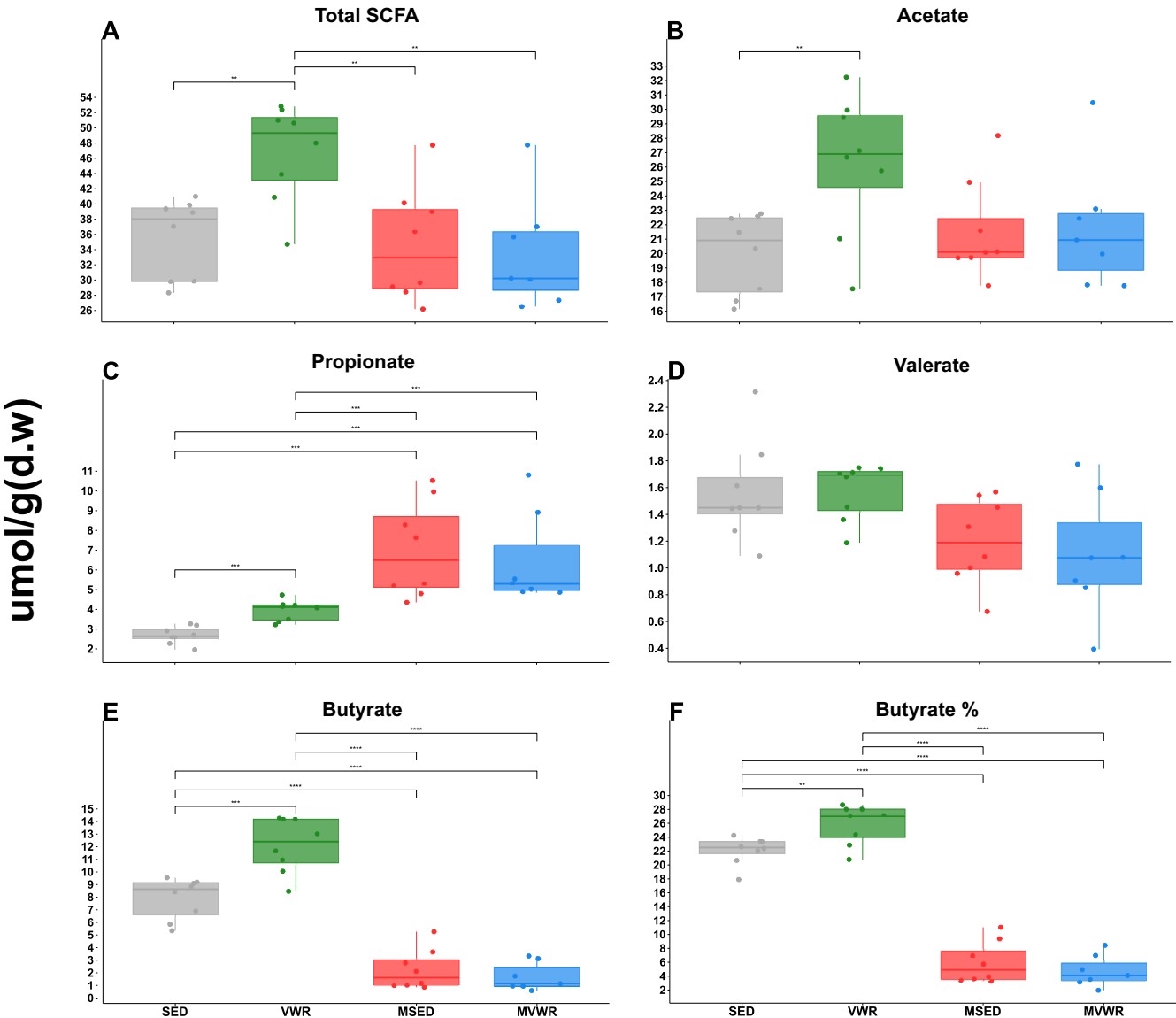

**FIG 3** Cecal SCFA composition. Cecal tissues were analyzed for SCFA composition using gas chromatography. The bottom and top boxes are the first and third quartiles, the middle bands inside the boxes are the medians, and the whiskers contain the upper and lower 1.5 interquartile range (IQR). An asterisk denotes significantly different pairs (adjusted $P < 0.05$). d.w, dry weight.

proportion of butyrate in VWR mice (~12%) was significantly higher than those in SED mice (~7.9%).

**Wheel running reduced gene expression of proinflammatory cytokines in WT but not *Muc2*$^{-/-}$ mice.** The exploratory principal component analysis (PCA) plot showed the clear separation of the WT versus *Muc2*$^{-/-}$ animals along the first principal component (PC1) axis, which accounted for 41.7% of the variation (Fig. S2). Further clustering between the SED and VWR groups, but not between MSED and MVWR, was observed along PC2, which accounted for an additional 17.7% of the variance. The result of the PERMANOVA test confirmed these observations, revealing a clear separation among groups (F, 10.513; $P < 0.01$). Pairwise comparison tests showed a statistically significant separation between all pairs except between MSED and MVWR. Complementary to these results, the generalized linear model (GLM) also showed a significant difference (adjusted $P = 0.001$) across groups. Notably, VWR mice had significantly lower tumor necrosis factor alpha (TNF-$\alpha$), transforming growth factor beta (TGF-$\beta$), interferon gamma (IFN-$\gamma$), and regenerating islet-derived protein 3 (RegIII-$\gamma$)

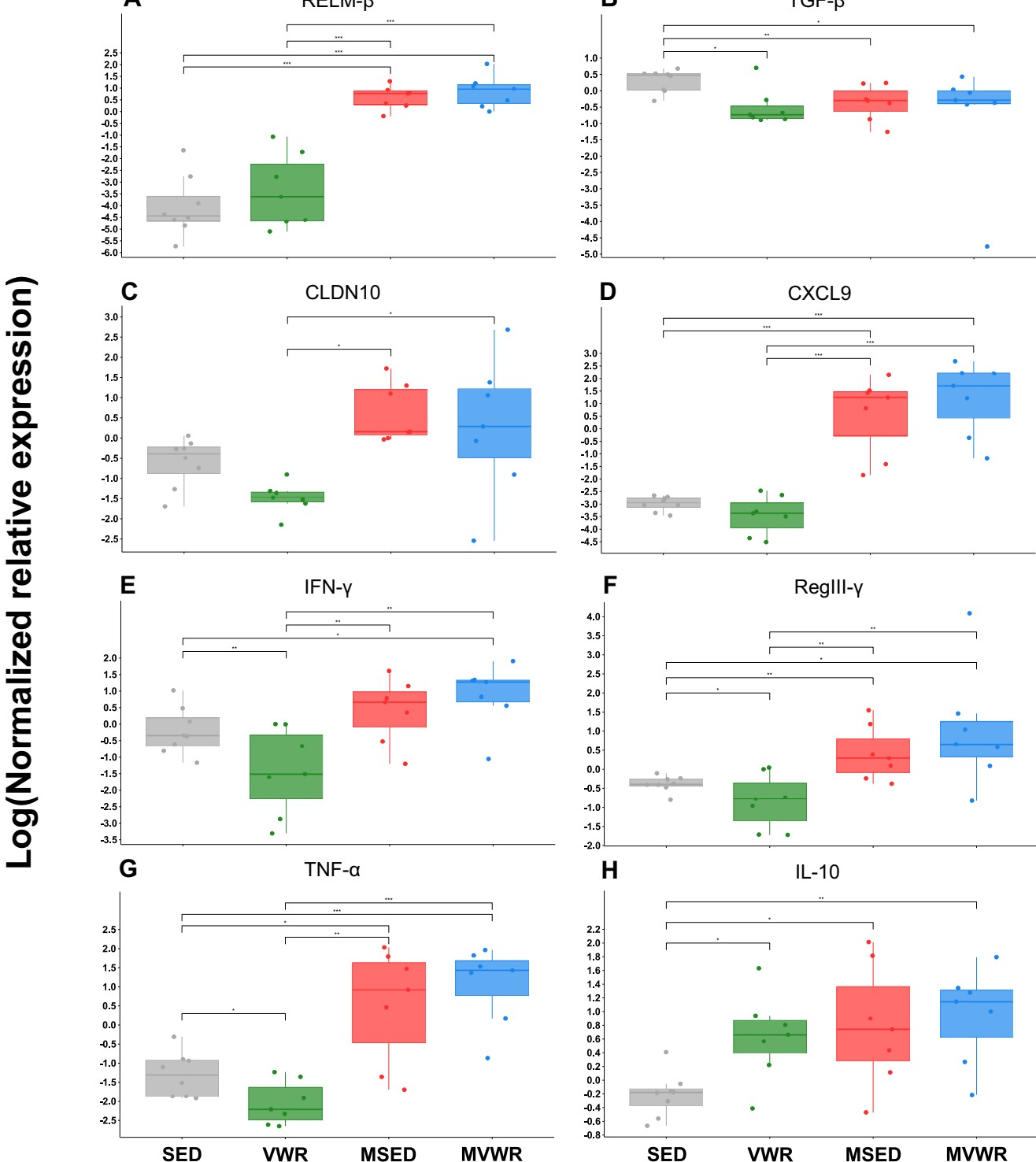

**FIG 4** Colonic mRNA gene expression. The relative mRNA gene expression of selected pro- and anti-inflammatory mediators in colon. The bottom and top of the boxes are the first and third quartiles, the middle bands inside the boxes are the medians, and the whiskers contain the upper and lower 1.5 interquartile range (IQR). An asterisk denotes significantly different pairs (adjusted $P < 0.05$).

levels than SED mice (Fig. 4); no changes were detected between MVWR and MSED animals. $Muc2^{-/-}$ animals had increased concentrations of interleukin-10 (IL-10), resistin-like molecule beta (RELM-$\beta$), CXCL9, RegIII-$\gamma$, and TNF-$\alpha$ compared to those of their WT counterparts.

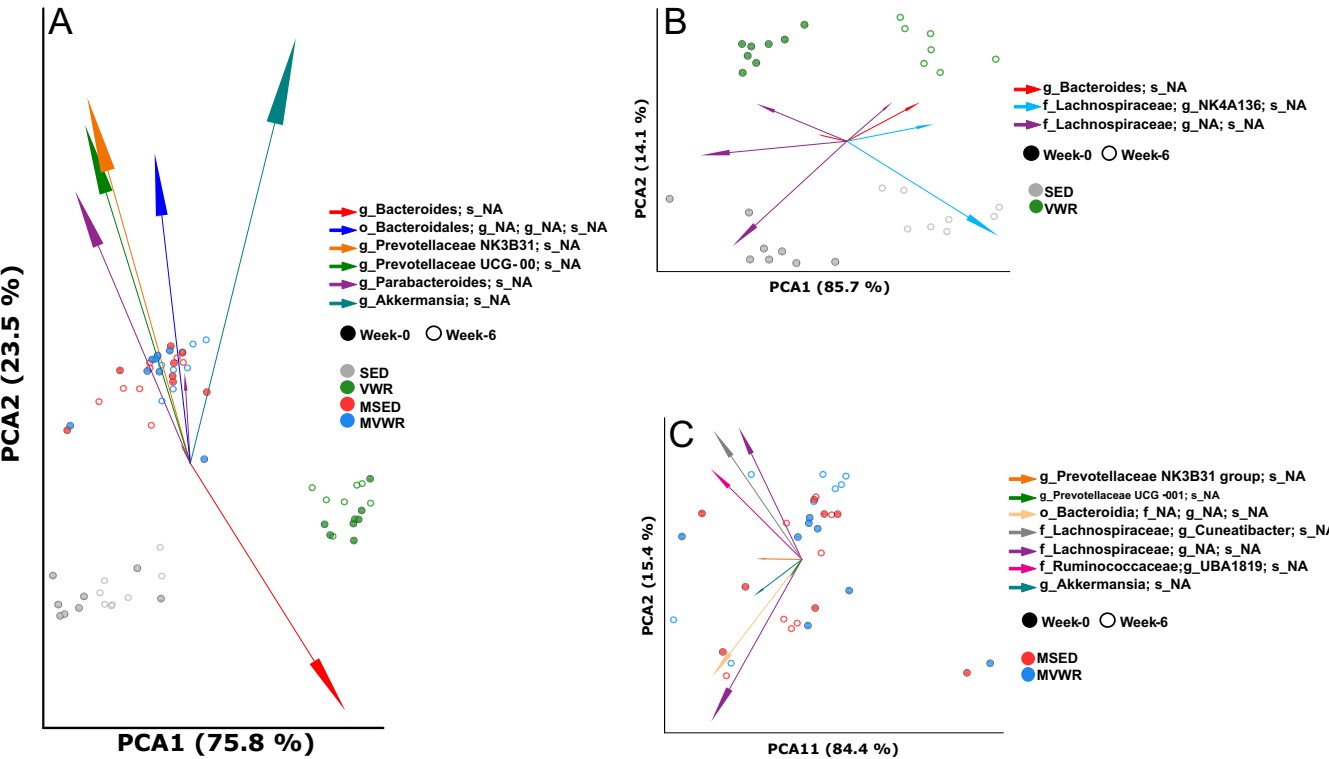

**FIG 5** Changes in fecal microbiome composition across time and physical activity. PCA biplot of robust Aitchison distances visualized where data points represent individual mice colored by their group designation, with spheres corresponding to samples at week 0 and rings representing week 6. The vectors represent the topmost significant ASV loadings driving differences in the ordination space. (A) While the *Muc2*$^{-/-}$ mice appear to cluster together regardless of activity status, the WT mice appear to be separated at week 0, suggesting the presence of a batch effect; therefore, further analyses are focused on the change across time in each group separately. (B) Significant differences were detected as a function of both time and activity in WT mice, as determined by PERMANOVA test (pseudo F = 75.2; *P* = 0.001; permutations = 999). The pairwise test showed differences between all groups (*P* < 0.001). (C) No group differences were detected in *Muc2*$^{-/-}$ mice, as determined by a PERMANOVA test (pseudo F = 0.47; *P* = 0.86; permutations = 999).

**Wheel running had a moderate effect on the microbiome of WT but not *Muc2*$^{-/-}$ mice. (i) Beta diversity.** The PERMANOVA test showed a significant difference between *Muc2*$^{-/-}$ and WT animals, corresponding to clear clustering observed between these groups on the PCA plots (Fig. 5A). Importantly, however, there was a significant distance between SED and VWR animal clusters prior to treatment assignment. This strongly suggests the presence of a batch effect in our experiment, likely explained by the fact that the VWR animals were purchased at different times than the other groups and their microbiome also was sequenced separately. As batch effects are a well-known issue in short-read sequencing experiments (17), differences across groups then are likely confounded by this. Therefore, to mitigate this effect, in all subsequent analyses, changes in microbiome are either compared only within the same group across time or the change within each group is compared to changes in other groups. Pairwise analysis of each group comparing their week 0 to week 6 profiles showed changes in overall microbiome variation in all animals across time. In *Muc2*$^{-/-}$ animals, these changes were nonuniform and did not follow a predictable pattern (Fig. 5C). WT animals, on the other hand, showed a clear and unidirectional change (clear clustering) in their overall microbiome (Fig. 5B). A significant shift in community structure of both VWR and SED animals based on the Aitchison distances was observed by week 6 (pseudo F = 75.2; *P* = 0.001; permutations = 999), and these changes were significantly different from each other. These shifts suggest a distinct change in the structure of the microbiome as a function of time as well as physical activity in WT animals. We next compared the magnitude of change (delta = $\beta_{\text{week 6}} - \beta_{\text{week 0}}$) across all groups and found no significant differences between them (Fig. S3).

**(ii) Alpha diversity.** The difference in change of alpha diversity estimates between week 6 and week 0 showed no significant change across any group (Fig. 6A to C). The WT

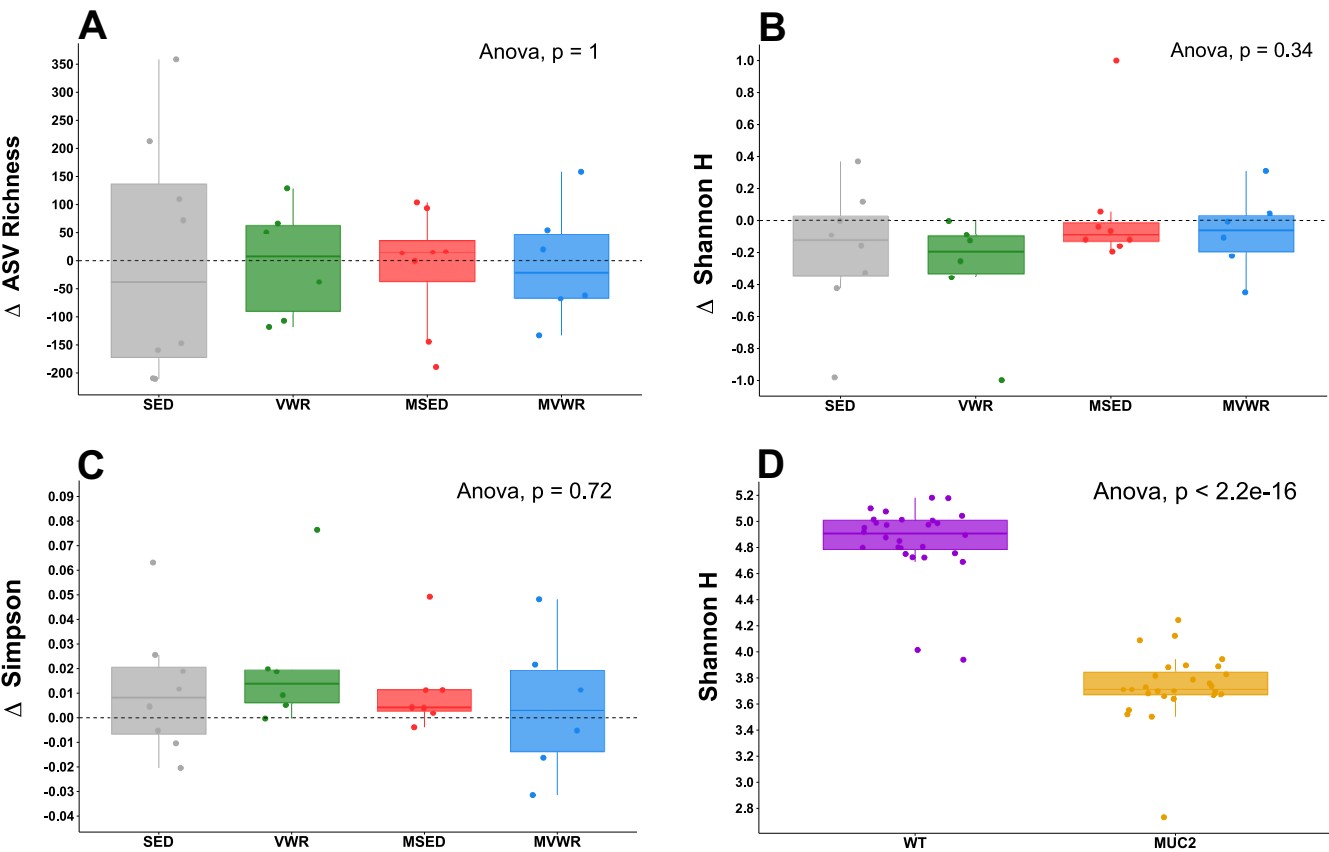

**FIG 6** Change in alpha diversity following 6 weeks of voluntary wheel running. Values are the differences between week 6 and week 0. In panels A to C, each group's change in alpha diversity after 6 weeks was compared to its own week 0 values as well as to those of other groups. No differences were observed across any groups. (D) The overall Shannon diversity of WT mice is significantly higher than those of $Muc2^{-/-}$ mice.

mice had a significantly higher ($P < 0.001$) overall diversity than $Muc2^{-/-}$ animals in all examined diversity indexes: species richness, Shannon (Fig. 6D), and Simpson index.

**(iii) Differential abundance testing.** Only WT animals showed statistically significant changes in relative abundance of individual taxa across time. In SED animals, 21 significant taxa were changed by week 6, and 20 were different in VWR animals (Fig. 7). Of these 41 total changed taxa, only 4 were common across both groups (Fig. S4). Interestingly, the relative abundance of these 4 taxa changed in the same direction, suggesting an age-driven effect. These changes were an increase in an ASV belonging to the genus *Ruminiclostridium5*, decrease in two ASVs from genera "*Negativibacillus*" and *Harryflintia*, and decrease in an ASV from the *Lachnospiraceae* family.

**(iv) Predicted phenotypic traits.** BugBase's prediction of each community's phenotypic traits suggests major differences between WT and $Muc2^{-/-}$ animals (Fig. S5). Bacterial communities in $Muc2^{-/-}$ mice were composed of significantly higher abundances of Gram-negative, aerobic, and facultative anaerobic bacteria with a higher potential for biofilm formation. Their communities also housed fewer bacteria with mobile elements and had an overall lowered tolerance for oxidative stress. At week 6, only VWR mice showed significant changes in their bacterial phenotypes compared to week 0. Their communities showed lower average abundances of mobility-containing (~12%, $P < 0.05$) and Gram-positive (~14%, $P < 0.001$) bacteria (Fig. 8).

## DISCUSSION

$Muc2^{-/-}$ **mice vastly differ from the WT in their colonic cytokine, SCFA, and microbial profiles.** $Muc2^{-/-}$ mice displayed clinical and histological symptoms of moderate colitis corresponding to the expected severity of this model at 11 weeks of

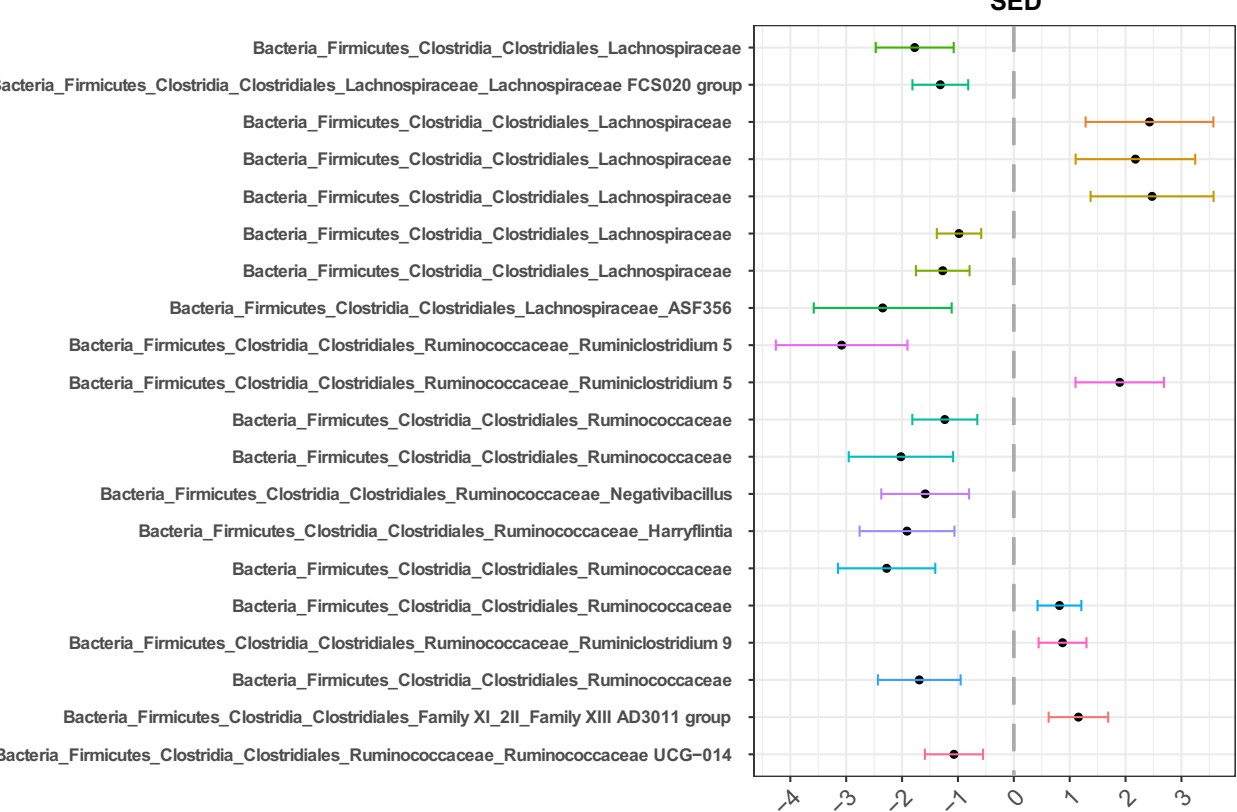

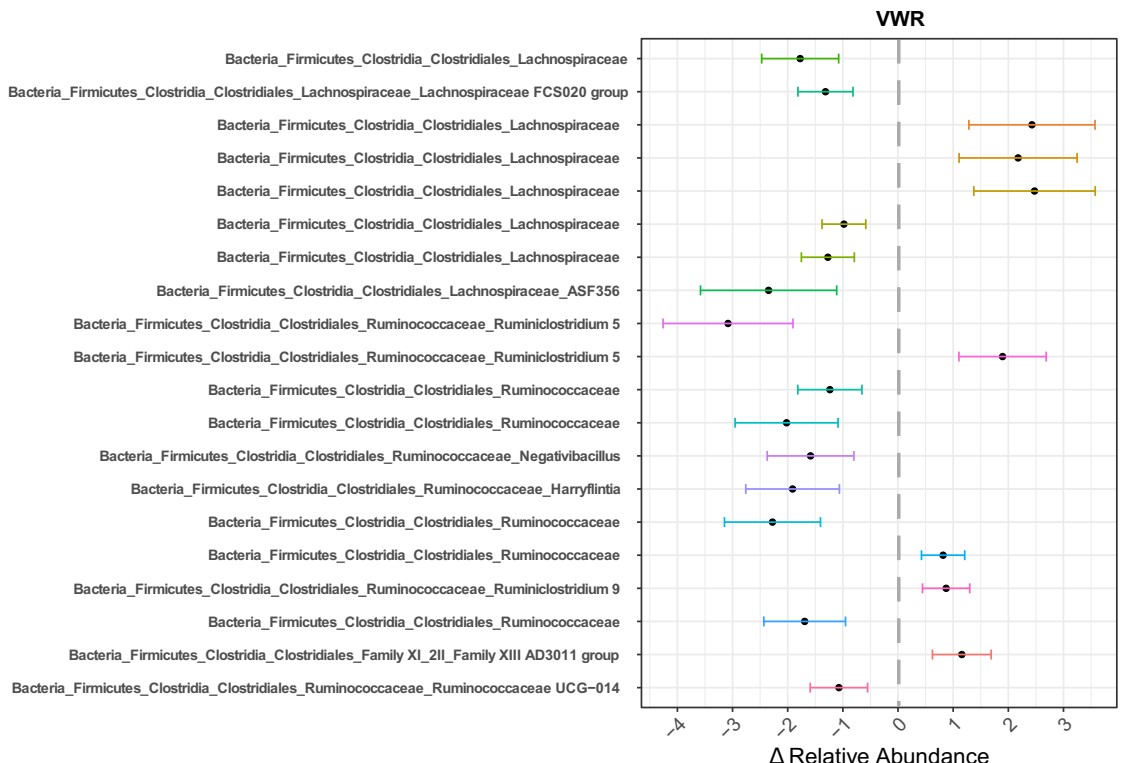

**FIG 7** Significant differences in fecal ASVs across time in WT mice. WT mice, but not *Muc2⁻/⁻* mice, showed a significant change in relative abundance of several ASVs (threshold set at FDR of $P < 0.01$) across time. Different ASVs are changed in VWR compared to SED mice. The points on the plot represent the mean change in relative abundances of each ASV at week 6 compared to week 0, and bars represent the 95% prediction intervals. No plot is shown for *Muc2⁻/⁻* mice, as there were no significant changes in any ASVs in these groups.

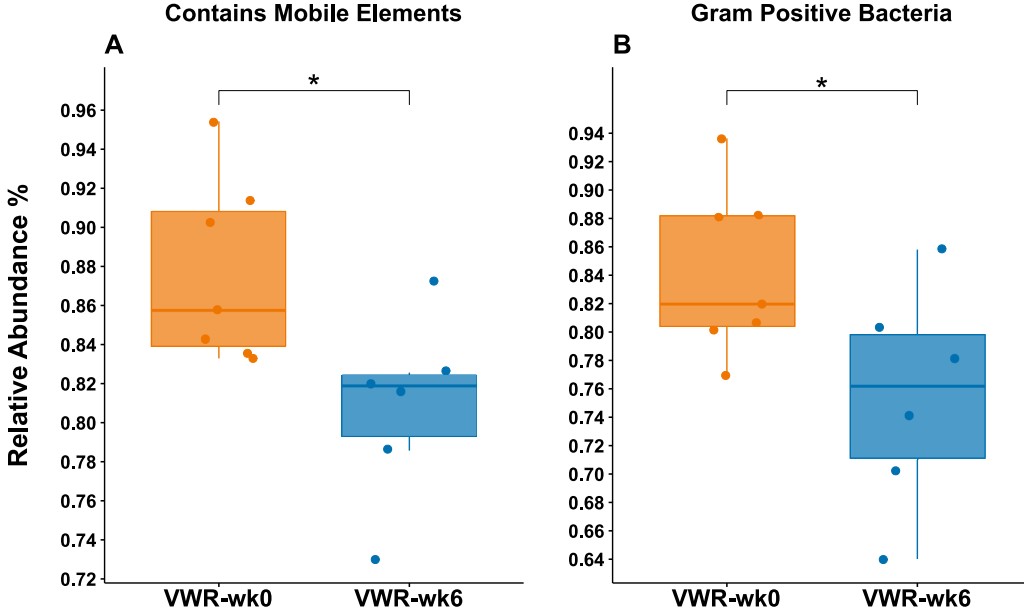

**FIG 8** Change in predicted phenotypic traits of microbial communities in VWR mice following 6 weeks of wheel running. BugBase was used to predict the composition of microbial communities based on their bacterial traits. Only VWR mice showed a statistically significant change in overall microbial community traits of those containing a mobile element (A) and Gram-positive bacteria (B) across time.

age in our facilities. The colonic gene expression of inflammatory cytokine TNF-$\alpha$ and the mucosal defense factor RELM-$\beta$, as well as the antimicrobial peptide RegIII-$\gamma$, was upregulated in $Muc2^{-/-}$ animals, as observed previously (15). Notably, the anti-inflammatory cytokine IL-10 was overexpressed in $Muc2^{-/-}$ compared to C57BL/6 WT mice. When in a healthy state, the expression of IL-10 may be associated with increased tolerance to inflammatory events. In $Muc2^{-/-}$ animals, this upregulation is essential to the host's efforts to suppress the excessive inflammation resulting from continuous exposure to bacterial ligands. Indeed, $Muc2^{-/-}$ IL-$10^{-/-}$ double knockout mice show highly exacerbated colitis clinical signs (18) compared to those of the deletion of either gene separately. The increase in IL-10 also was previously observed in chemical models of colitis (9, 11). We further detected significant overexpression of CXCL9 in $Muc2^{-/-}$ animals. CXCL9 is a chemokine involved in regulating leukocyte trafficking, likely in response to the exposure of bacterial ligands to host cells. CXCL9 overexpression also has been reported in IBD patients (19). Overall, the cytokine profile of $Muc2^{-/-}$ animals reflects those expected in human IBD.

$Muc2^{-/-}$ mice born without a normal mucus layer house drastically less diverse and different bacterial communities than WT mice, as evidenced by the clear clustering of this group from WT animals in our PCA plots. Similar to patterns seen in IBD patients (20) or chemically induced murine colitis (21, 22), $Muc2^{-/-}$ animals had an overall reduced $\alpha$-diversity compared to their WT counterparts. The dominant taxa in WT mice were generally of the *Bacteroides* genus, *Clostridiales* order, and *Lachnospiraceae* family, while $Muc2^{-/-}$ animals were dominated by members of the *Muribaculaceae* family (formerly known as S24-7 [23]) and the *Akkermansia* genus of the *Verrucomicrobia* phylum (see Fig. S7 in the supplemental material). While our taxonomic classifier was unable to confidently differentiate between the 2 sole species (*Akkermansia muciniphila* and *Akkermansia glycaniphila*) within this genus, it is reasonable to assume that the observed taxa were *A. muciniphila* isolates, as *A. glycanphilia* has, to date, been isolated only from python feces (24). *A. muciniphila* is perhaps the most surprising finding in this group, as this species is known, and named, for its ability to degrade mucin and is broadly considered a beneficial bacterium in a variety of chronic diseases, including IBD

(25–27). The broader implications of this finding are beyond the scope of the present study. However, it does warrant the reassessment of the characterization of *A. mucini-phila* as a mucin-loving species to one that thrives in the absence of mucin. The bacterial phenotypic traits of *Muc2*$^{-/-}$ animals were predicted to be higher in abundances of Gram-positive, aerobic, and biofilm-forming groups than in WT mice. Lastly, the cecal SCFA of *Muc2*$^{-/-}$ mice were composed of significantly less butyrate and higher propionate concentrations than SED animals. The increased propionate levels in these animals are likely associated with the high abundances of *A. muciniphila*, a prominent propionate producer (28, 29). Overall, we found the *Muc2*$^{-/-}$ model of colitis to capture many components of human IBD, especially those with impaired mucosal integrity.

**Wheel running in *Muc2*$^{-/-}$ mice does not reduce the severity of chronic colitis.** Contrary to our primary hypothesis, we found that 6 weeks of wheel running in *Muc2*$^{-/-}$ mice did not improve the severity of clinical signs, histopathological scores, colonic expression of inflammatory cytokines, or abundances of cecal SCFAs and did not alter the gut microbial composition in a consistent manner. These findings contrast with others that show protective effects of VWR or forced treadmill running in chemically induced models of colitis (8–11). The fundamental difference between those studies and ours is in the model of colitis used. Previously, VWR was initiated in healthy animals prior to disease induction with chemical toxins, whereas in our study, wheel running is imposed over an existing disease state as a therapeutic intervention. This suggests that PA prior to disease onset primes various components of intestinal health, enhancing its tolerance to injury. The effects of PA following disease onset, on the other hand, either are abolished or are overwhelmed by stronger disease signaling. It is also possible that the physiological benefits of PA depend on the presence of a healthy mucosal layer. This is supported by our findings that wheel running in WT but not *Muc2*$^{-/-}$ animals leads to significantly lower levels of proinflammatory colonic cytokines, increased anti-inflammatory IL-10, and increased levels of beneficial SCFAs. Given that UC patients typically have defective and thinning colonic mucosal layers, an exercise prescription in these populations may have limited direct benefits on their intestinal health. However, the well-documented benefits of exercise are instituted across various other sites and systems of the body, which may indirectly result in improving primary and secondary disease symptoms through other pathways not accounted for in this experiment. For example, PA as a primary intervention has been associated with improved quality of life in IBD patients (30) and inversely correlated with loss of bone mass density, a common risk factor in this population (31, 32).

**VWR significantly attenuates proinflammatory, and upregulates anti-inflammatory, cytokines in WT mice.** Compared to SED animals, the wheel-running mice showed lower levels of inflammatory cytokines TNF-$\alpha$, IFN-$\gamma$, and TGF-$\beta$, all of which have been implicated in IBD (33). TNF-$\alpha$ is perhaps the most studied cytokine in relation to IBD, as it plays a crucial role in innate and adaptive immunity and is directly involved in apoptotic processes in the intestines (34). It is found in significantly higher abundances in IBD patients (35) as well as murine colitis (36), making its regulation an obvious target for disease management. TNF-$\alpha$ inhibition using monoclonal antibodies is the most common target of biological therapies for moderate to severe IBD. The role of IFN-$\gamma$ in colitis pathogenesis is less consistent across the literature; however, its overproduction has been shown in CD (37, 38) and UC patients (39). In dextran-sodium sulfate (DSS)-induced colitis models, neutralization antibodies against IFN-$\gamma$ significantly reduced disease severity (40), while IFN-$\gamma^{-/-}$ mice were completely protected from disease clinical signs (41). Anti-IFN-$\gamma$ antibody treatments in human IBD are less effective, however, with their efficacy being dependent on baseline C-reactive protein levels (42), highlighting the need for treatment personalization. TGF-$\beta$ is a pleiotropic cytokine that is ubiquitously produced by many cells and is involved in various immune functions, including both anti- and proinflammatory actions. These include the suppression of immune responses through the recruitment of Tregs, which, in turn, produce IL-10, but TGF-$\beta$ can also elicit potent Th17 responses to combat extracellular

bacteria (43). TGF-$\beta$ is found in higher concentrations in intestines of IBD patients (44, 45) due to increased exposure of microbial ligands to host epithelial cells. Inversely, the attenuated levels of this cytokine in the VWR animals may reflect a decrease in bacterial antigen exposure to the IEC, suggesting reduced levels of host-microbe interactions in the mucosa. Alternatively, reduced TGF-$\beta$ also indicates reduced Treg activity in VWR mice; however, the increase in Treg-derived IL-10 in these animals does not support this notion. IL-10 is an anti-inflammatory cytokine ubiquitously secreted by Tregs and is the primary driver of immunosuppressant actions in the intestines. Polymorphism in IL-10 promoters has been linked to IBD, making IL-10 supplementation a potential target for IBD therapy; however, clinical studies of IL-10 therapy to date have not been significantly effective (46). The significant increase in IL-10 in VWR mice suggests higher Treg activity, which is associated with reduced inflammation. This is in agreement with others who showed a significant increase in murine intestinal IL-10 following treadmill running or swimming (47, 48). However, it is unclear whether this reflects a beneficial increase in anti-inflammatory events or simply an adaptive response to changes in the microbial composition. Gram-negative bacteria preferentially stimulate IL-10 production and are associated with higher virulence due to increases in abundance of lipopolysaccharides bound to their cell walls (49). Thus, the higher expression of IL-10 in VWR animals is likely correlated with increased abundance of Gram-negative bacteria observed in these mice. Further investigations are needed to determine the consequence of these changes. Taken together, the reduction of these proinflammatory cytokines and increase in anti-inflammatory IL-10 in VWR animals suggests a primed anti-inflammatory state in healthy WT but not diseased intestines, marking them as potentially important targets for prevention and remission maintenance therapy.

**VWR significantly augments SCFA content in WT but not $Muc2^{-/-}$ mice.** SCFAs are metabolic by-products of bacterial fermentation of dietary fibers in the colon and are involved in various physiological processes of the host. Aberrant intestinal SCFA content has been implicated in various diseases, such as irritable bowel syndrome, cardiovascular disease, certain cancer types, and IBD (50–52). The most abundant of these, acetate, propionate, and butyrate, which make up over >95% of SCFA in humans (53), are markedly decreased in IBD patients (54), while their exogenous delivery can reduce inflammation via inhibition of TNF-$\alpha$ release from neutrophils (51, 55). Overall, increases in these SCFA, especially butyrate, appear to positively influence IBD (56). We found an overall higher abundance of total cecal SCFA, acetate, butyrate, and propionate in response to wheel running in WT but not $Muc2^{-/-}$ animals. This is in accordance with others showing higher butyrate concentrations in wheel-running rats (57), following exercise training in lean humans (58) and elite athletes (59). We have also previously observed a positive association between higher butyrate levels and VO$_2$ peak in healthy humans (16). The increase in these SCFA may simply reflect higher energy demands of colonocytes, which utilize SCFA as their primary energy substrate. Interestingly, when we analyzed SCFA content in relative abundances, we saw a significant increase in the relative abundance of butyrate but not acetate or propionate. This suggests a preference in VWR animals for the production of butyrate and its accompanying anti-inflammatory properties. These findings further support the patterns of anti-inflammatory priming we observe in these animals, contributing to an overall healthier intestinal environment following physical activity. The mechanisms behind PA-induced changes in SCFA are not known; however, given that SCFA are primarily produced by the intestinal microbiota, it is highly likely that changes in SCFA are linked to the observed changes in the microbiome. SCFA affects microbiota dynamics, as they are directly involved in chemical balance and pH regulation of the intestines (60) and, in turn, the microbiota can affect SCFA production and use, establishing a bidirectional affiliation.

**Wheel running has limited but significant effects on the intestinal bacterial composition of WT but not $Muc2^{-/-}$ mice.** In this study, neither time nor wheel running had any effect on any alpha diversity metrics measured. The effects of PA on

alpha diversity is not consistent in the literature. For example, the findings here are in contrast to our own previous observations in healthy humans that showed a significant correlation between alpha diversity and cardiorespiratory fitness (16). Others have also reported that elite athletes have higher alpha diversity than sedentary controls (61) or that exercise training in mice leads to increased Shannon diversity (62, 63). However, in agreement with the current experiment, PA has been shown to have no effect on alpha diversity in mice (64–66), rats (67), and humans (58, 68). The reasons for these discrepancies are not clear, although multiple factors, such as differences in animal vendors and facilities, DNA extraction methods and sequencing, bioinformatics analysis, and statistical testing methods, are likely involved. Additionally, one particularly important consideration in comparing animal models of PA is the total volume of activity performed. For example, across the aforementioned studies utilizing wheel running in mice, we noticed a wide range (~2.5 to 10 km) of average daily running distances reported. Different volumes of PA are likely to elicit different physiological responses, which can extend to the microbiome.

Comparisons of the across-sample diversity (beta diversity) in *Muc2*$^{-/-}$ animals showed no patterns of change as a function of time or wheel running. In WT animals, however, the change in Aitchison distances between weeks 6 and 0 were significantly different between VWR and SED groups, indicating both time and wheel running are important factors in the observed shift in community composition. However, the magnitude of this change across time between the groups was not different.

Univariate analyses of individual taxa in *Muc2*$^{-/-}$ animals also showed no significant changes in any microbial clades across either group. A lack of significant change in these animals suggests that the presence of a healthy mucosal layer is required to mediate PA-induced changes in community composition in the colon. In WT animals, the relative abundances of over 20 taxa in each group were significantly different by week 6, with only four of those taxa common to both VWR and SED groups. The changing taxa in either group belonged primarily to the *Ruminococcaceae* and *Lachnospiraceae* families, with some species increasing while others decreased. Notably, in SED animals, 5 species from the *Bacteroides* genus were decreased by week 6, while no changes in this genus were detected in VWR animals. Species within the *Bacteroides* genus are Gram-negative, obligate anaerobes that are among the most abundant found within the mammalian intestine, and they carry important functions, such as the breaking down of complex glycans, refining the gut environment by reducing intracellular oxygen levels, and preventing the colonization of opportunistic pathogens (69). *Bacteroides* spp. are also one of the primary propionate producers in the mammalian gut (29); therefore, the observed reduction in members of this group in SED animals may, in part, explain their lower propionate levels compared to those of VWR mice. Given that we were unable to classify these ASVs to the species level, it is difficult to speculate further on the biological implications of these observations. Perhaps the most studied species in this group, *Bacteroides fragilis*, has been shown to be protective against DSS colitis in mice by stimulating IL-10 expression (70, 71). Given that we also observed significantly lower IL-10 expression in SED mice, it is reasonable to speculate that at least one of the unclassified *Bacteroides* ASVs in this group is *B. fragilis* and is linked to the relatively lower expression of this protective cytokine in those animals.

Analyzing the bacterial consortia based on their predicted phenotypic traits revealed additional information regarding the effect of wheel running on the overall community. Following wheel running, WT, but not *Muc2*$^{-/-}$, mice had a significant reduction (~14%) in the total abundance of Gram-positive bacteria. This is supported by the observed decreases in several members of the Gram-positive *Ruminococcaceae* in these mice as well as the attenuated expression of RegIII-γ, an antimicrobial peptide that specifically targets the surface peptidoglycan layer of Gram-positive bacteria. The implications of this phenotypic shift in microbiota of healthy individuals is not known but may provide a clue for understanding the adaptations of the intestinal environment to the physiological stresses of PA. Furthermore, mirroring the shift in Gram-positive phenotype was the decreased relative abundances of bacteria containing mobile

elements. These refer to microevolutionary processes, such as transposons, i.e., segments of DNA with the ability to move locations within the genome, and bacterial plasmids, which are involved in horizontal gene transfer. These events are typically associated with the sharing of virulence factors between bacterial cells and increased resistance to antibiotics. The higher abundances of mobile elements in bacteria from these mice is likely not indicative of antibiotic resistance but rather associated with higher abundances of Gram-negative bacteria representing more mobile elements. The results of these predictions should be interpreted with caution, however, as these mobile elements can rapidly become population specific within an individual, precluding inference across similar experimental groups (72).

**Summary.** In contrast to our hypothesis, we found that 6 weeks of wheel running did not ameliorate any clinical signs of colitis in $Muc2^{-/-}$ animals, and it did not influence any components of the intestinal environment, such as the expression of various cytokines and the production of SCFA. Wheel running in healthy WT C57BL/6 mice, on the other hand, imposed various physiological effects on the gut, including the downregulation of proinflammatory and upregulation of anti-inflammatory cytokine gene expression, and increased concentration of total SCFA, including butyrate, acetate, and propionate. Wheel running further led to a shift in bacterial community structure corresponding to higher abundances of Gram-negative bacteria. As these physiological changes have been associated with protection against chronic inflammatory diseases in humans, such as IBD, we conclude that PA prior to disease onset can prime the intestines, enhancing their tolerance to inflammation. These benefits, however, are lost when PA is imposed in the absence of a healthy mucosal layer. The results here do not capture any possible sex-driven effects, which should be considered in future work. Overall, the findings here suggest that PA in healthy individuals is an important preventative against intestinal diseases such as IBD.

## MATERIALS AND METHODS

**Experimental procedures.** To test our primary hypothesis that habitual physical activity can be protective during IBD as a treatment rather than a preventative therapy, as has been previously shown, we chose to utilize $Muc2^{-/-}$ mice as a lifelong model of murine colitis. In our facility, $Muc2^{-/-}$ mice are underweight at birth and display early signs of colitis that are mild to moderate in severity up to ~3 months of age, at which point they accelerate rapidly, reaching high severity by 4 months. Therefore, we designed our experiment to conclude prior to the 3-month time point under the speculation that higher severity of disease symptoms would preclude the animals from voluntarily running on wheels. Following weaning at 5 weeks of age, animals were randomly assigned to individual cages under one of four groups ($n = 8$ per group): wild-type (WT) C57BL/6 mice with access to a free running wheel (VWR) or a locked wheel (SED) and $Muc2^{-/-}$ mice with access to a free wheel (MVWR) or a locked wheel (MSED). We chose wheel running as a model of PA over forced exercise, as mice voluntarily run higher total distances on free wheels than when forced on a treadmill (73). Forced exercise can also cause significant stress in rodents (74) and has been shown to exacerbate colitis severity in C57BL/6 mice (9). A 6-week VWR intervention period was selected based on previous reports showing this to be sufficient in eliciting significant protection against chemically induced models of colitis (75, 76). The primary responses of interest during this period were weekly clinical signs of colitis and histopathological disease scores at terminus. Additional responses of interest were changes to intestinal microbial composition and function, immunity, and SCFA production in response to PA. The variables associated with these additional responses are described in detail below. By examining these variables, we aimed to elucidate the mechanisms by which PA recruits protective actions.

**Animals.** All procedures involving the care and handling of the mice were approved by the UBC Committee on Animal Care, under the guidelines of the Canadian Council on Animal Care. Four-week-old male WT C57BL/6 mice were purchased from Charles River (Vancouver, CA) and kept under specific-pathogen-free conditions. $Muc2^{-/-}$ mice, also generated on a C57BL/6 genetic background, were bred in-house, with the founding colonies kindly donated by Bruce Vallance from the Child and Family Research Institute (UBC Vancouver). We chose male mice in the current experiment, as females exhibit greater variability in wheel-running behavior, which has been linked to the estrus cycle (77). All animals were housed in a temperature-controlled room (22 ± 2°C) on a 12-h light/dark cycle with access to acidified water and irradiated food (PicoLab Rodent Diet 20–5053; Quebec, CA) *ad libitum*. The assignment of mice to experimental groups was carried out using a random number generator immediately preceding individual cage allocation.

**Voluntary wheel running and food and water intake.** The running wheels (diameter, 10.16 cm; width, 5.1 cm; Columbus Instruments) were mounted to the top of the cage lids and were programmed to record the total number of revolutions at 1-h intervals for the duration of the experiment. Body weights, food consumption, and water intake were measured weekly at approximately the same time

mSystems®

during the light cycle. Food weight measurements consisted of subtracting the week's remaining pellets on the cage lids and bottoms from that week's starting weight.

**Tissue collection.** For fecal sample collection, mice were kept briefly in isolation in sterile and DNAzap-treated containers until defecation. Collected fecal pellets, which were used for microbiome surveying, were immediately snap-frozen in liquid nitrogen and then stored at $-80°C$ until further analyses. Fecal samples were collected on day 1 immediately following assignment to individual cages and again on the final experiment day immediately preceding tissue collection. Animals were euthanized by cervical dislocation while under deep isoflurane anesthesia. The cecum was isolated, its content removed, and tissue frozen in liquid nitrogen for further analyses of SCFA composition. Colon tissues were collected in the following manner: starting from distal end, two consecutive $\sim$1.5-cm sections were collected, with the most distal section being fixed in 10% neutral buffered formalin for histopathology and the proximal section stored in RNAlater (Thermo Fisher Scientific) for use in cytokine gene expression assays. All frozen samples were then stored at $-80°C$ until further use.

**Clinical and histopathological scoring.** Disease progression and severity in *Muc2*⁻/⁻ animals was assessed based on an in-house clinical signs scoring system and represented by a variable we call the "disease score." Briefly, each animal was graded weekly based on the observed behavior from a distance, stool/rectal bleeding, stool consistency, weight loss, and hydration, with each variable being assigned a score of 0 to 4. The humane endpoint was set as a total cumulative score of $\geq$12, rectal prolapse, or a loss of >20% body weight for 2 consecutive days. No animals reached a humane endpoint in this study.

For histopathological scoring, colon cross sections were fixed in 10% neutral buffered formalin at 4°C overnight, washed 3 times with phosphate-buffered saline (PBS; pH 7.4), transferred to 70% ethanol, and sent for dehydration, paraffin embedding, sectioning, and hematoxylin and eosin (H&E) staining at Wax-it Histology Services (Vancouver, Canada). Tissue slides were coded throughout the microscopy analyses, and the two investigators scoring the histopathology were blinded to the groupings. H&E-stained sections were viewed under $\times$200 magnification on an Olympus IX81 microscope and the full image stitched together using MetaMorph software. Stitched images were imported into ImageJ, version 1.51r (78), for scoring. Disease severity in colonic cross sections from the *Muc2*⁻/⁻ animals were assessed using a previously described scoring system (79). In brief, a total score was calculated for each mouse using the following criteria. (i) Edema, compared to a healthy WT control, was given the following scores: 0, no change; 1, mild (<10%); 2, moderate (10 to 40%); 3, profound (>40%). (ii) Epithelial hyperplasia was scored as the average height of crypts as a percentage above the height of a healthy control: 0, no change; 1, 1 to 50%; 2, 51 to 100%; 3, >100%. (iii) Epithelial integrity was determined as the shedding and shape of the epithelial layer compared to that of the healthy control: 0, no change; 1, <10 epithelial cells shedding per lesion; 2, 11 to 20 epithelial cells shedding per lesion; 3, epithelial ulceration; 4, epithelial ulceration with severe crypt destruction. (iv) Cell infiltration was measured as the presence of immune cells in submucosa: 0, none; 1, mild (2 to 43); 2, moderate (44 to 86); 3, severe (87 to 217). The resulting histopathological score had a maximum value of 13.

**Reverse transcriptase-qPCR.** To identify the potential immunological pathways involved in PA-derived protection, we examined the gene expression of several key immune markers commonly associated with colitis. mRNA gene expression for tumor necrosis factor alpha (TNF-$\alpha$), interferon gamma (IFN-$\gamma$), resistin-like molecule beta (RELM-$\beta$), regenerating islet-derived protein 3 (RegIII-$\gamma$), transforming growth factor beta (TGF-$\beta$), chemokine C-X-C motif ligand 9 (Cxcl9), and claudin 10 (Cldn10) was measured in colon tissues. Total RNA was purified from tissues using Qiagen RNeasy kits according to the manufacturer's instructions, with an additional initial bead-beating step (30 s three times at 30 Hz) on a Retsch MixerMill MM 400 homogenizer. cDNA then was synthesized using the iScript cDNA synthesis kit (Bio-Rad) in 10-$\mu$l reaction mixtures. The RNA and cDNA products' purity and quantity were assessed by a NanoDrop spectrophotometer (Thermo Scientific). The cDNA products were normalized to $\sim$40 ng/$\mu$l with DNase-free sterile water prior to quantitative PCRs (qPCRs).

A 10-$\mu$l reverse transcriptase (RT)-qPCR consisted of 0.2 $\mu$l of each forward and reverse primer (10 mM), 5 $\mu$l of Sso Fast Eva Green Supermix (Bio-Rad), 3.6 $\mu$l DNase-free water, and 1 $\mu$l of cDNA template. Reactions were run in triplicates using the Bio-Rad CFX96 Touch thermocycler and analyzed using Bio-Rad CFX Maestro software 1.1 (v4.1). The median quantitation cycle (Cq) value from each sample was used to calculate the $2^{-\Delta\Delta CT}$ value based on the reference gene TATA box binding protein (Tbp). A list of all the primer sets, melting temperatures, efficiencies, and the detailed thermocycler protocol used in this study are described in Fig. S6 in the supplemental material.

**Short-chain fatty acids.** SCFA, a by-product of microbial fermentation, are an essential component of a healthy gut environment (reviewed in reference 53). They not only serve as a primary food source for the colonocytes but also have immunogenic properties that, in concert with the host immunity, are integral in maintaining gut homeostasis. We previously showed that in healthy humans, cardiorespiratory fitness was positively correlated with fecal butyrate (16), an SCFA with known anti-inflammatory properties in the gut (80). Therefore, we hypothesized that SCFA profiles of VWR mice would differ from those of SED mice, favoring the production of beneficial butyrate that may be involved in protection against colitis. Therefore, we analyzed SCFA (acetic, propionic, heptanoic, valeric, caproic, and butyric acids) in cecal tissues by gas chromatography (GC) as described previously (81). In brief, $\sim$50 mg of stool was homogenized with isopropyl alcohol, containing 2-ethylbutyric acid at 0.01% (vol/vol) as an internal standard, at 30 Hz for 13 min using metal beads. Homogenates were centrifuged twice, and the cleared supernatant was injected into a Trace 1300 gas chromatograph equipped with a flame-ionization detector with an AI1310 autosampler (Thermo Fisher Scientific) in splitless mode. Data were processed using Chromeleon 7 software. Half of the cecal tissue was freeze-dried to measure the dry weight, and measurements are expressed as micromoles per gram of dry weight.

**DNA extraction and 16S rRNA amplicon preparation.** The effects of PA on the intestinal micro-biome have recently arisen as an area of great interest (reviewed in reference 82). To date, several studies in mice have shown significant changes to the microbiome associated with either VWR or forced treadmill running (64, 75, 83, 84). Allen et al. (58) further showed that transplanting the microbiome of exercised mice into germfree mice conferred protection against dextran-sodium sulfate (DSS)-induced colitis, highlighting the importance of PA-derived changes in the microbiome. To examine such potential changes in the gut microbiome, we surveyed the fecal microbiome of our mice using high-throughput sequencing. DNA was extracted from fecal samples using the QIAmp DNA stool minikit (Qiagen) according to the manufacturer's instructions after three 30-s beat beatings as before. Amplicon libraries were prepared according to the Illumina 16S metagenomic sequencing library preparation manual. In brief, the V3-V4 hypervariable region of the 16S bacterial rRNA gene was amplified using recommended 341F and 805R degenerate primer sets, which create an amplicon of ∼460 bp. Amplicons were purified using AMPure XP beads and adapters, and dual-index barcodes (Nextera XT) were attached to the amplicons to facilitate multiplex sequencing. Following a secondary cleanup step, libraries were quality controlled on an Experion automated electrophoresis system (Bio-Rad) and sent to The Applied Genomic Core (TAGC) facility at the University of Alberta (Edmonton, Canada), where they were normalized using the fluorometric method (Qubit, Thermo Fisher Scientific) and sequenced using the Illumina MiSeq platform with a V3 reagent kit, allowing for two 300-bp cycles.

**Bioinformatics.** All bioinformatics processes were performed using a combination of R statistical software (85) and the QIIME 2 platform (86) using the various built-in plugins described below. Demultiplexed sequences were obtained from the sequencing facility and primers removed using cutadapt (87). Sequences then underwent quality filtering, dereplication, denoising, merging, and chimera removal using DADA2 (88). The output of this process is a feature table of ASVs that is a higher-resolution analogue of traditional operational taxonomic unit (OTU) tables. To aid in the removal of nonspecific host contaminants, a positive filter was applied to all reads using the latest available Greengenes (13_8) (89) database (clustered at 88% identity). All ASVs were searched against the reference reads using VSEARCH (90), and any that did not match the reference sequences at a minimum of 70% identity at 70% alignment were discarded. For analyses encompassing phylogenetic information, a phylogenetic tree was constructed using a SATé-enabled phylogenetic placement (SEPP) technique as implemented in the q2-fragment-insertion plugin (91) using a backbone tree build based on the SILVA (128) database (92). Taxonomic classification of the ASVs was carried out using IDTAXA (93). It has been proposed that the functional repertoire of the gut microbiota is more sensitive to perturbation than taxonomic changes and, therefore, is crucial in identifying underlying physiological signals (94). To predict the functional potential and phenotype of the microbiome, we used BugBase (95), which utilizes PICRUSt's (96) extended ancestral-state reconstruction algorithm for metagenome composition prediction. As these tools require sequences to be classified against the Greengenes taxonomy assignments, we used VSEARCH to pick closed-reference OTUs from our denoised feature table at a 97% similarity threshold against the 99% identity clustered Greengenes database.

**Statistical analyses.** All statistical analyses were performed using R, version 3.5.1, unless stated otherwise. During the third week of the experiment, the VWR animals were unintentionally exposed to 3 days of irregular light-dark cycles as a result of an electrical malfunction with the lighting in the animal room. While the exact nature of this disruption is not known, the wheel-running data during this period suggests a period of reduced activity. The issue was resolved by the third day, and the animals did not display any signs of stress or irregular behavior; therefore, we consider this to be of minimal impact to the experiment. However, as a precaution, we chose to analyze the data as a $4 \times 1$ (groups) factorial design rather than $2 \times 2$ (activity $\times$ genotype), as we could not definitively eliminate the possibility that wheel running in this group was impacted by the brief interruption.

**Wheel running.** To determine whether WT and $Muc2^{-/-}$ mice ran similar distances throughout the experiment, we first analyzed total weekly distances (kilometers) run by each group across the 6 weeks using linear mixed-effects regression (LMER) using the lme4 package, with individual animals set as the random effect and groups as the fixed effects. Homoscedasticity and linearity of the models were assessed using diagnostic plots of the residuals.

**Body weights and food/water intake.** To monitor overall behavioral changes of mice as a result of PA between WT and $Muc2^{-/-}$ mice, we examined weekly body weights and food and water intake across the 6 weeks. To account for natural differences in starting body weights, total weight gained relative to starting body weights was calculated each week. Body weight and food and water intake across the 6 weeks were each assessed separately using a repeated-measures LMER with time coded as a random effect and groups as a fixed effect. A Tukey's honestly significant different *post hoc* test with the Benjamini-Hochberg (BH) *P* adjustment method was used when an overall significance (set as $P < 0.05$) in the models was detected.

**Clinical and histopathological scoring.** We used a cumulative link model (CLM) with a logit link to evaluate whether the disease score, and, separately, the histopathological score differed among treatment groups. This proportional odds-type test is more appropriate for ordinal data than classic linear regressions. For clinical scores, the model included time and groups as the fixed effects and individual animal ID as the random effect. For histopathological score, the total average scores of the MSED and MVWR groups were separately analyzed using the same method but without the time random effect. We implemented the analyses using the ordinal R package.

**Colon mRNA gene expression.** To test whether the expression of colonic mRNA genes differed across groups, we first explored the overall abundance of all surveyed genes simultaneously using an ordination method. The Euclidean distances of Hellinger-transformed relative gene expression values

were ordinated onto a principal component analysis (PCA) plot. When a clear clustering was observed based on group assignments, differences in variance across these groups were assessed using a permutational multivariate analysis of variance (PERMANOVA) test using the vegan package, and pairwise differences calculated using pairwiseAdonis with BH adjustment for multiple testing. For differential abundance testing of each cytokine, a multivariable generalized linear model (GLM) test was carried out using the mvabund package (97). This fits separate GLMs to each cytokine while accounting for nonindependence and adjusting for multiple testing. The negative binomial distribution assumption was selected for the model, and the mean variance plot was used to assess the model fit. A Kruskal-Wallis *post hoc* test was carried out on individual genes when significance was detected in the overall model. Pairwise comparisons across groups were carried out using Conover's test for multiple comparisons within the PMCMRplus package.

**Short-chain fatty acids.** Similar to the cytokine data analysis, to test the differences in abundance of SCFA across groups, concentrations of various cecal SCFA were assessed using a multi-GLM test. *Post hoc* tests were carried out on individual SCFA identified as significant from the univariate results from the global model.

**Microbial analysis.** Following our previous observation in humans that showed distinct microbial community characteristics and metagenomic functions associated with higher cardiorespiratory fitness, we evaluated whether similar patterns emerged in mice. Community structural patterns of fecal bacteria across samples ($\beta$ diversity) were explored using the q2-DEICODE plugin (98). DEICODE is a compositionally aware method that utilizes a form of robust Aitchison distances to create a species abundance distance matrix of ASVs that then can be projected onto a PCA biplot. We visualized this using the Emperor interactive graphic tool (99). To reveal possible group differences, a PERMANOVA (100) test was conducted on all groups across time. Pairwise testing was then monitored using a Kruskal-Wallis test with a BH adjustment to control for the false discovery rate (FDR).

The overall within-sample diversity ($\alpha$ diversity) for each sample was estimated based on the species richness and the Simpson index and Shannon diversity using the DivNet package (101). For each group, as well as other groups, the difference between a sample's week 6 and week 0 diversity score was calculated and used to determine whether those changes differed from zero (Wilcoxon test) (ANOVA).

Differential abundance testing of individual taxa was performed using the CornCob package (102).

BugBase was used to determine high-level phenotypes of bacterial communities based on the following default traits: Gram-negative versus Gram-positive, biofilm forming, mobile element containing, oxidative stress tolerance, pathogenic potential, and oxygen utilizing. Pre- and posttreatment differences in relative abundances of these elements were tested in each group using a Kruskal-Wallis test with Benjamini-Hochberg adjustment of $P$ values to control the FDR.

**Data availability.** All data and codes used for all aspects of this work, including all metadata, histology figures, and raw sequence files, are currently available publicly at https://osf.io/9awgd/. Additionally, raw FASTQ files are also available through Qiita (https://qiita.ucsd.edu/) study ID 13321 and EBI (accession no. ERP123717).

All used software packages, versions, and parameters from the QIIME 2 software are available under the "provenance" tab of the accompanying qiime2 zip artifact (.qza) files. These files can be viewed locally or on a web browser at https://view.qiime2.org/.

## SUPPLEMENTAL MATERIAL

Supplemental material is available online only.

**FIG S1**, EPS file, 0.1 MB.
**FIG S2**, EPS file, 0.1 MB.
**FIG S3**, EPS file, 0.1 MB.
**FIG S4**, EPS file, 0.2 MB.
**FIG S5**, EPS file, 0.2 MB.
**FIG S6**, EPS file, 0.1 MB.
**FIG S7**, EPS file, 0.2 MB.

## ACKNOWLEDGMENTS

Authors made the following contributions: conceptualization, M.E. and D.L.G.; data curation, M.E.; formal analysis, M.E. and J.P.; funding acquisition, D.L.G.; investigation, M.E., D.W.M., C.Q., A.G., S.K.G., and J.A.B.; methodology, M.E. and D.L.G.; project administration, M.E. and D.L.G.; resources, D.L.G.; software, M.E. and J.P.; supervision, D.L.G.; validation, M.E. and J.P.; visualization, M.E.; writing (original draft), M.E.; writing (review and editing), M.E., D.W.M., C.Q., J.P., J.B., S.K.G., and D.L.G.; design, S.G.; analysis, S.G.; expertise, S.G.; approval of the final manuscript, S.G.

This project was funded by Crohns & Colitis Canada (CCC) and Natural Sciences and Engineering Research Council (NSERC), awarded to D.G. M.E. was funded by an NSERC PGSD. C.Q. was funded by CIHR CGSD. J.P. was funded by NSERC discovery. J.B. was funded by NSERC USRA. S.G. was a Michael Smith Foundation for Health Research

Scholar funded by a Canadian Association of Diabetes Grant in Aid and an NSERC Discovery Grant.

**Note Added after Publication**

After original publication of this paper, several changes were required and have been made in this version of the article.

S. Ghosh was added to the article byline.

The following was added to the end of the first paragraph of Acknowledgments: "...design, S.G.; analysis, S.G.; expertise, S.G.; approval of the final manuscript, S.G."

The following was added to the end of the second paragraph of Acknowledgments: "S.G. was a Michael Smith Foundation for Health Research Scholar funded by a Canadian Association of Diabetes Grant in Aid and an NSERC Discovery Grant."

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
