## [Reviewer comments · mSystems]

Physical activity shapes the intestinal microbiome and immunity of healthy mice but has no protective effects against colitis in MUC2^{-/-} mice

Mehrbod Estaki, Douglas Morck, Candice Quin, Jason Pither, Jacqueline Barnett, Sandeep Gill, and Deanna Gibson

Corresponding Author(s): Deanna Gibson, University of British Columbia

Review Timeline:

Submission Date:	June 16, 2020
Editorial Decision:	July 23, 2020
Revision Received:	August 29, 2020
Accepted:	September 16, 2020

Editor: Morgan Langille

Reviewer(s): The reviewers have opted to remain anonymous.

Transaction Report:

DOI: <https://doi.org/10.1128/mSystems.00515-20>

July 23, 2020

Dr. Deanna L Gibson
University of British Columbia
Kelowna
Canada

Re: mSystems00515-20 (Physical activity shapes the intestinal microbiome and immunity of healthy mice but has no protective effects against colitis in MUC2^{-/-} mice)

Dear Dr. Deanna L Gibson:

Your manuscript has been thoroughly reviewed by two reviewers and I concur with their comments that only minor modifications are required in order for it to be acceptable for publication. Please prepare a revised version of your manuscript including a response to each of the reviewers comments.

Below you will find the comments of the reviewers.

To submit your modified manuscript, log onto the eJP submission site at <https://msystems.msubmit.net/cgi-bin/main.plex>. If you cannot remember your password, click the "Can't remember your password?" link and follow the instructions on the screen. Go to Author Tasks and click the appropriate manuscript title to begin the resubmission process. The information that you entered when you first submitted the paper will be displayed. Please update the information as necessary. Provide (1) point-by-point responses to the issues raised by the reviewers as file type "Response to Reviewers," not in your cover letter, and (2) a PDF file that indicates the changes from the original submission (by highlighting or underlining the changes) as file type "Marked Up Manuscript - For Review Only."

Due to the SARS-CoV-2 pandemic, our typical 60 day deadline for revisions will not be applied. I hope that you will be able to submit a revised manuscript soon, but want to reassure you that the journal will be flexible in terms of timing, particularly if experimental revisions are needed. When you are ready to resubmit, please know that our staff and Editors are working remotely and handling submissions without delay. If you do not wish to modify the manuscript and prefer to submit it to another journal, please notify me of your decision immediately so that the manuscript may be formally withdrawn from consideration by mSystems.

To avoid unnecessary delay in publication should your modified manuscript be accepted, it is important that all elements you upload meet the technical requirements for production. I strongly recommend that you check your digital images using the Rapid Inspector tool at <http://rapidinspector.cadmus.com/RapidInspector/zmw/>.

Sincerely,

Morgan Langille

Editor, mSystems

Journals Department
Reviewer comments:

Reviewer #1 (Comments for the Author):

The manuscript by Estaki et al. describes the role of exercise in modifying the gut microbiota in wild-type (WT) and mice with a deficient mucus barrier (MUC2^{-/-}). The authors show that while exercise alters the gut microbiota and metabolites in WT mice (a phenomenon previously described by others), such changes fail to occur in MUC2^{-/-}. These results are important and novel, as they uncover a key role of the mucus layer in regulating exercise-induced microbiome changes. In addition MUC2^{-/-} mice are prone to colitis onset and hyper inflammatory states in the colon. This is important as previous reports have shown exercise to be effective in limiting chemical-induced colitis severity and reducing colonic inflammation. However, here the authors show here that MUC2^{-/-} mice exhibit no such benefit from exercise, suggesting that exercise may limit colitis severity though changes in mucus.

Overall, the paper is well written, well organized and brings up important considerations for the field. Nevertheless, there are some minor issues with the manuscript that should be addressed prior to publication.

- 1) I would recommend combining Figure 1 and Figure 2. It would improve the flow of the manuscript as body weight is a clinical outcome that can be viewed alongside histopathology.
- 2) Please make headers of sections more descriptive of the results. For instance, instead of the header on line 382 which currently reads "Colon mRNA gene expression". The header could read something like "Exercise reduces expression of inflammatory genes in WT but not MUC2^{-/-} mice".
- 3) Please avoid statistical jargon like this in lines 363-365 (see especially underlined sections). "The results of the GLM indicated a significant group effect (Dev: 14.83, P<0.01) and the univariate tests showed significant differences in acetate, propionate, butyrate, and valerate across groups. The results of the post-hoc analyses on these SCFAs.
- 4) In figure 7, why is microbiome results of MUC2^{-/-} animals not shown? These should be reported.

Reviewer #2 (Comments for the Author):

This is an interesting and substantive contribution to the literature. Novel information regarding the potential benefits of regular exercise in the context of the absence of a normal gut mucus layer are included. The experiment has the proper controls and all there is rationale for all the variables measured.

There are some issues that dampen enthusiasm.

-what is the rationale for only including male mice?

-state why acidified water was used

-verify that histology and observation of clinical symptoms were assessed in a blinded to treatment manner? Also, was there more than one rater? If so, what was inter-rater reliability?

-comment on the very low N for analysis of microbiome data

-why was there a batch effect? Mice should have been randomized to treatment from the same batched shipment of mice to control for batch effects. This should be discussed in the discussion (e.g. limitations) of experiment.

RE: Physical activity shapes the intestinal microbiome and immunity of healthy mice but has no protective effects against colitis in Muc2^{-/-} mice

We are very grateful for the insightful comments from both reviewers and are happy that they share our enthusiasm for this project. We have addressed, to the best of our ability, their comments and concerns. Please find below a point-by-point summary of our reply and corresponding changes within the manuscript. In addition, we have made some minor grammatical edits to the manuscript, included a Qiita study # for the DNA sequences (EBI submission ongoing), as well as made some purely aesthetic changes to the main Figures and Supplementary Material (for example increasing axis text sizes to improve readability). We look forward to working with the publishing team to move this manuscript to the next step.

Reviewer #1

Query 1- I would recommend combining Figure 1 and Figure 2. It would improve the flow of the manuscript as body weight is a clinical outcome that can be viewed alongside histopathology.

Response: We considered combining the figures but in the end, we still believe that we should keep the figures separate for the following reasons:

Unlike Figure 1, Figure 2 is focused on Muc2^{-/-} animals only, and is meant to highlight the response to wheel running on disease outcomes which excludes WT animals. Additionally, the clinical disease scores (Figure 2A) already takes into account weight loss as a component of its accumulated score. As so we deemed it redundant to provide weight-loss data in two formats in the same panel. We'll leave the final decision to the editor and are happy to revisit this on further request.

Query 2- Please make headers of sections more descriptive of the results. For instance, instead of the header on line 382 which currently reads "Colon mRNA gene expression". The header could read something like "Exercise reduces expression of inflammatory genes in WT but not Muc2^{-/-} mice".

Response: We agree with this excellent recommendation and have made this change.

Query 3 - Please avoid statistical jargon like this in lines 363-365 (see especially underlined sections). "The results of the GLM indicated a significant group effect (Dev: 14.83, P<0.01) and the univariate tests showed significant differences in acetate, propionate, butyrate, and valerate across groups. The results of the post-hoc analyses on these SCFAs.

Response: This comment is well received and we reduced statistical jargon throughout the results section, replaced some test-specific terms with more commonly used terms, and removed some redundant lines.

Query 4- In figure 7, why is microbiome results of Muc2^{-/-} animals not shown? These should be reported.

Response: As we have described in the results section: “Only WT animals showed statistically significant changes in relative abundance of individual taxa across time”, therefore as there were no statistically different ASVs in Muc2^{-/-} animals, there was nothing to plot. We have included an additional line explaining this in the Figure 7 caption.

Reviewer #2

Query 1-what is the rationale for only including male mice?

Response: Thank you for this excellent question. Admittedly one of the limitations of the current study is that we cannot capture sex-specific effects. We chose male mice in the current experiment as females exhibit greater variability in wheel running which has been linked to the estrus cycle. For example: female mice run much greater distances after the pro-estrus phase and less after the diestrus (PMID: 19139751). We initially speculated that the amount of wheel running may differently alter the microbiome and so opted to use male mice to reduce this potential confounding factor. In a related study (not yet published) we have indeed found that the amount of wheel running can impact the change observed in the microbiome. We have included an explanation to this effect in the manuscript under both the methods section and discussion.

Query 2 -state why acidified water was used

Response: The use of acidified water is a very common standard practice in many animal facilities, including ours, it is used as a means to control certain pathogens such as *Pseudomonas* species. We have previously shown (<https://www.tandfonline.com/doi/full/10.1080/19490976.2018.1539599>) that the type of water used in mouse studies can significantly alter intestinal microbiome and thus should be reported.

Query 3 -verify that histology and observation of clinical symptoms were assessed in a blinded to treatment manner?

Response: Yes they were blinded for histology scoring, as described in the Methods > Clinical and Histopathological Scoring section: “*Tissue slides were coded throughout the microscopy analyses and investigators scoring histopathology were blinded to the groupings.*”

Only 1 person per session assessed the animals for clinical signs and they were not blinded to the groupings. This was not possible because a) the Muc2^{-/-} animals are visibly different in weight, behavior, and their cages are often smeared with blood and diarrhea; and b) as these animals require additional monitoring by our technicians due to their colitis, their cage labelling, although coded, was also revealing as needed for the independent animal welfare assessments for compliance.

Query 4-Also, was there more than one rater?

Response: Yes, two blinded investigators scored the histology slides. This has been clarified in the methodology section.

Query 5-If so, what was inter-rater reliability?

Response: The scores from the 2 investigators were highly in agreement. Of the 16 Muc2^{-/-} animals scored, only 4 of them were not exactly rated the same between the 2 individuals. In these 4 cases, the variability was very low, with the max difference being .5 (of a total score of 13). Note that we have included the raw H&E images as well as our scores onto the public OSF folder linked in the manuscript, should independent evaluation be needed.

Query 6--comment on the very low N for analysis of microbiome data

Response: Power calculations based on a previous pilot study had suggested that this sample size was sufficient to capture microbial signals. During data analysis, we removed one animal from each wheel running group (VWR, MVWR) because they did not run on the wheels thus were considered biological outliers.

We further lost 1 mouse from the VWR group due to technical issues at the sequencing facility.

Query 7-why was there a batch effect? Mice should have been randomized to treatment from the same batched shipment of mice to control for batch effects. This should be discussed in the discussion (e.g. limitations) of experiment.

Response: Thank you for this query as we agree this is a valid point. We initially had designed our experiment exactly as suggested above, mice from the same time and batch were randomly assigned to either SED or VWR groups, however, following the first arm of our experiment we noticed that all of our VWR mice had run very little on the wheels. We later discovered that the wheels had not been properly installed, as the axles on them were too tight and too difficult for mice to run freely on. We therefore had to repeat the VWR arm of the project, so we purchased a new batch and re-ran them separately. We believe this to be the root cause of the observed batch effect. We do describe this in the manuscript and additionally discuss our steps to mitigate the influence of batch effect, ex:

“Therefore, to mitigate this effect, in all subsequent analyses, changes in microbiome are either only compared within the same group across time, or the change within each group is compared to changes in other groups.”

We believe that this approach, while limiting to some types of statistical analyses, minimizes the effects of the batch effect.

Sincerely, on behalf of the co-authors,

Mehrbod Estaki
Deanna L. Gibson

September 16, 2020

Dr. Deanna L Gibson
University of British Columbia
Kelowna
Canada

Re: mSystems00515-20R1 (Physical activity shapes the intestinal microbiome and immunity of healthy mice but has no protective effects against colitis in MUC2-/- mice)

Dear Dr. Deanna L Gibson:

Your manuscript has been accepted, and I am forwarding it to the ASM Journals Department for publication. For your reference, ASM Journals' address is given below. Before it can be scheduled for publication, your manuscript will be checked by the mSystems senior production editor, Ellie Ghatineh, to make sure that all elements meet the technical requirements for publication. She will contact you if anything needs to be revised before copyediting and production can begin. Otherwise, you will be notified when your proofs are ready to be viewed.

Sincerely,

Morgan Langille
Editor, mSystems

Journals Department
Fig. S6: Accept

Fig. S5: Accept

Fig. S1: Accept

Fig. S7: Accept

Fig. S2: Accept

Fig S4: Accept

Fig. S3: Accept